# A complex epistatic network limits the mutational reversibility in the influenza hemagglutinin receptor-binding site

Nicholas C. Wu [1], Andrew J. Thompson [2], Jia Xie[3], Chih-Wei Lin[3], Corwin M. Nycholat[2], Xueyong Zhu[1], Richard A. Lerner[3,4], James C. Paulson [2,5] & Ian A. Wilson [1,4]

The hemagglutinin (HA) receptor-binding site (RBS) in human influenza A viruses is critical for attachment to host cells, which imposes a functional constraint on its natural evolution. On the other hand, being part of the major antigenic sites, the HA RBS of human H3N2 viruses needs to constantly mutate to evade the immune system. From large-scale mutagenesis experiments, we here show that several of the natural RBS substitutions become integrated into an extensive epistatic network that prevents substitution reversion. X-ray structural analysis reveals the mechanistic consequences as well as changes in the mode of receptor binding. Further studies are necessary to elucidate whether such entrenchment limits future options for immune escape or adversely affect long-term viral fitness.

---

[1] Department of Integrative Structural and Computational Biology, The Scripps Research Institute, La Jolla, CA 92037, USA. [2] Department of Molecular Medicine, The Scripps Research Institute, La Jolla, CA 92037, USA. [3] Department of Chemistry, The Scripps Research Institute, La Jolla, CA 92037, USA. [4] The Skaggs Institute for Chemical Biology, The Scripps Research Institute, La Jolla, CA 92037, USA. [5] Department of Immunology and Microbiology, The Scripps Research Institute, La Jolla, CA 92037, USA. Correspondence and requests for materials should be addressed to I.A.W. (email: wilson@scripps.edu)

Since entering the human population in 1968, H3N2 influenza A virus has evolved extensively, especially in its hemagglutinin (HA), due to selection pressure from the human immune system[1]. Substitutions acquired continuously over time from immune evasion have led to extensive antigenic drift throughout the past half-century[2–4]. The HA glycoprotein on the viral surface is responsible for cell entry and is the main focus of the humoral immune response. The hypervariable globular head domain is composed entirely of HA1 residues and contains the receptor-binding site (RBS), whereas the more conserved elongated stem, assembled from HA2 and the N- and C- terminal regions of HA1, houses the membrane fusion machinery. The HA RBS is a shallow pocket on the apex of the HA1 globular head[5], and is formed from four structural elements: the 130-loop, 150-loop, 190-helix, and 220-loop, named after their location on the HA primary sequence. The HA RBS engages host receptor sialylated glycans and consequently is relatively conserved in sequence compared to the rest of HA1.

In avian H1N1 and H3N2 influenza A viruses, the amino acids at residues 190, 225, 226, and 228 are Glu, Gly, Gln, and Gly, respectively, which are associated with the preference for α2–3 linked sialosides. For avian viruses to transmit in humans, the receptor specificity has to switch to α2–6 linked sialosides. Such a change in tropism of the H3N2 subtype occurred in 1968 through a double substitution Q226L/G228S within the HA RBS, whereas a different double substitution, E190D/G225D, was employed by the H1N1 subtype in 1918 (refs. [6–8]). While E190D and G225D substitutions were thought to be irrelevant in human adaptation of the H3N2 subtype, recent human H3N2 strains have acquired these human H1N1 signature RBS mutations in addition to other substitutions within the HA RBS, likely as a result of antigenic drift to escape from the immune system[2,3,9,10].

Here we show that reverting Asp at residue 190 in recent human H3N2 strains back to a Glu, the ancestral state for human H3N2 viruses, significantly impaired viral replication. A series of deep mutational scanning and virus rescue experiments based on reverting H3N2 A/Brisbane/10/2007 (Bris07) to sequences present in the RBS of earlier strains revealed that the subsequent entrenchment of Asp at residue 190 was attributable to a number of other substitutions that then arose within and surrounding the HA RBS. Crystal structure analysis further indicated that the positioning of the 190-helix and receptor binding have changed during the evolution of human H3N2 viruses. Overall, our work elucidates the importance of intragenic epistasis and its complexity in the HA RBS in human H3N2 viruses, and of the potential consequences for escape from immune pressure.

## Results

### The loss of E190D reversibility in human H3N2 viruses.
Residue 190 was a glutamic acid residue (Glu) before the 1992–1993 influenza season and an aspartic acid (Asp) in almost all strains that emerged afterwards (Fig. 1a and Supplementary Fig. 1a). We performed a pilot virus rescue experiment by reverting HA residue 190 of a recent strain A/Victoria/361/2011 (Vic11) to the ancestor state, i.e. from Asp to Glu (D190E). Surprisingly, D190E in Vic11 significantly attenuated the virus (Supplementary Fig. 1b). In contrast, substituting residue 190 in the ancestor strain A/Hong Kong/1/1968 (HK68) to the descendant state, i.e. from Glu to Asp (E190D), did not alter virus replication fitness (Supplementary Fig. 1b). This mutational fitness effect is unlikely due to the difference in neuraminidase (NA), as swapping the NA between HK68 and Vic11 did not impose any fitness difference in virus replication (Supplementary Fig. 1b). This result suggests that the E190D reversibility is lost in recent human H3N2 strains. In fact, natural reversion is not very common in human H3N2

HA (Fig. 1b). While one intuitive explanation is that immune memory in the human population disfavors reversion, our observation here suggests that functional constraints may also be involved.

Next, we aimed to narrow down the time period when the reversibility of E190D was lost by employing eight strains from 1993 to 2011, namely A/Shangdong/9/1993 (Shang93), A/Moscow/10/1999 (Mos99), A/Wyoming/3/2003 (Wy03), A/Brisbane/10/2007 (Bris07), A/Perth/16/2009 (Perth09), Vic11, A/Michigan/15/2014 (Mich14), and A/North Dakota/26/2016 (NDako16). All eight strains carry an Asp at residue 190 (Supplementary Table 1) and their HA1 amino-acid sequences are shown in Supplementary Fig. 2. The fitness effect of D190E reversion in each of these nine strains was examined by virus rescue experiments. D190E imposed a significant fitness cost in Bris07, Perth09, Vic11, Mich14, and NDako16, but not in Shang93, Mos99, and Wy03 (Fig. 1c). This result implies that the reversibility of E190D was lost in early to mid-2000s. In fact, entrenchment or irreversibility is an important evolutionary consequence of epistasis[11–13]. Our subsequent analyses aimed to identify and characterize crucial epistatic interactions related to this observed entrenchment.

### Mapping the genetic determinants of E190D reversibility.
Reversion has been demonstrated as an effective approach to study evolutionary entrenchment[12]. To delineate the loss in E190D reversibility, we aimed to identify substitutions that could restore its reversibility in recent strains. In the panel of the eight strains tested, Bris07 was the earliest strain that lost the E190D reversibility and Wy03 was the latest strain that retained the E190D reversibility. Subsequently, we searched for substitutions that could restore E190D reversibility by studying reversions from Bris07 to Wy03. We focused on eight candidate substitutions that represented the reversion from Bris07 to Wy03 within or immediately surrounding the RBS, namely N145K, G186V, N189S, F193S, P194L, S219F, N225D, and P227S (Fig. 2a). To probe for the fitness effects of all possible combinations of those eight substitutions in Bris07 wild type (WT) and D190E genetic backgrounds ($n = 2^8 \times 2 = 512$ variants), a deep mutational scanning experiment was performed. Deep mutational scanning, which combines saturation mutagenesis and next-generation sequencing to quantify a large number of genetic variants in parallel[14], has previously been successfully used to study influenza virus HA[15–17]. The results from our deep mutational scanning experiment here were strongly correlated (Pearson correlation = 0.91–0.92) among three biological replicates (Supplementary Fig. 3a). The enrichment level of each variant from the plasmid mutant library to the post-selection mutant library was quantified by a relative fitness (RF) index (see Methods). The variant with the highest RF index that contained Glu190 was G186V/D190E/F193S/P194L/P227S (the top variant along the y-axis in Fig. 2b). In fact, G186V/F193S/P194L/P227S combination was also found in the second and third top variants that contained Glu190. This result suggests that multiple natural substitutions account for the loss of E190D reversibility.

### E190D reversibility is controlled by multiple substitutions.
We then performed a series of virus rescue experiments with individually constructed Bris07 mutants to corroborate the findings from the deep mutational scanning (Fig. 2c). G186V/F193S/P194L/P227S, which was the top variant from the deep mutational scanning experiment, compensated for the D190E reversion to produce a WT-like titer in the virus rescue experiment. We also found that G186V/F193S/P194L achieved a similar compensatory effect without P227S. Interestingly, none of

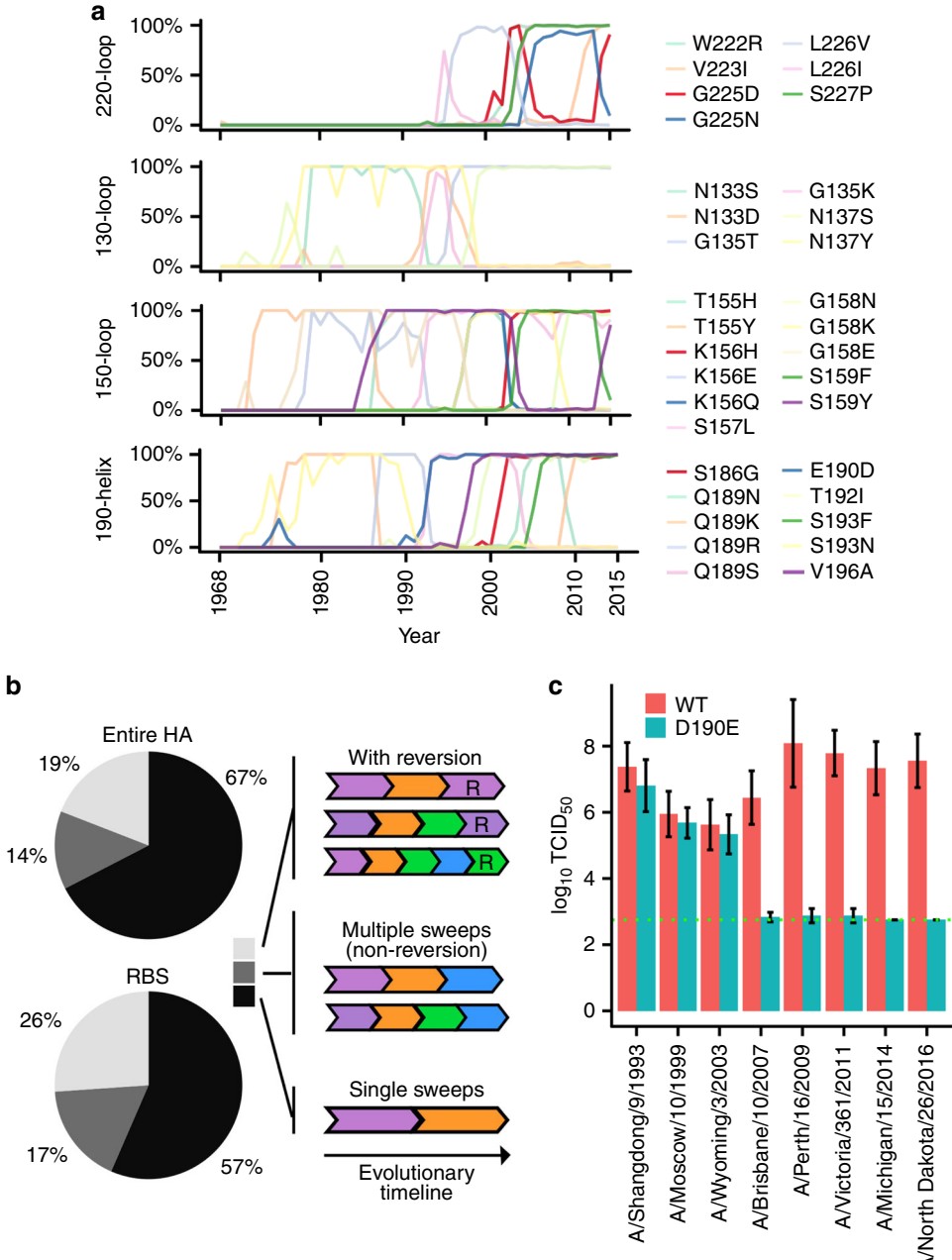

**Fig. 1** Evolution of HA RBS and the mutability of residue 190. **a** The frequencies of several representative substitutions in HA RBS in naturally circulating human H3N2 strains are shown. **b** Residues that are involved in selective sweeps during the natural evolution of human H3N2 viruses were analyzed for the entire HA and for the RBS, and were further classified into "with reversion", "multiple sweep", and "single sweep" (see Methods). The cartoon arrows on the right demonstrate several scenarios for the classification scheme. The evolutionary timeline of a given residue is shown from the left to right, where different colors represent different dominant amino-acid variants at the given residue. "R" indicates the occurrence of reversion. **c** Effect of D190E on viral replication fitness for different chimeric H3N2 strains was examined by virus rescue experiment. Virus titer was measured by $TCID_{50}$. Error bars indicate the standard deviation of three independent experiments. The green dashed line represents the lower detection limit

G186V, F193S, and P194L substitutions alone could compensate D190E. We then aimed to test the compensatory effects of these two sets of substitutions on other recent H3 strains (Perth09, Vic11, Mich14, and NDako16). P194L was not applicable to those H3 strains because they already possessed a Leu at residue 194 (Supplementary Table 1). Pro194 in Bris07 is likely a result of egg-passaging during clinical isolation[18–20]. G186V/F193S/P227S and G186V/F193S were indeed able to compensate D190E in Perth09, Vic11, Mich14, and NDako16 (Supplementary Fig. 3b), suggesting their compensatory effects are not strain-specific.

When we tested those eight reversions of interest from Bris07 to Wy03 one by one, N145K and P227S partially compensated D190E (Fig. 2c). We further introduced N225G into D190E/ P227S and observed a WT-like titer. N225G was selected because of the prevalence of Gly at residue 225 in H3 strains before the early 2000s and its functional association with residue 190 for switching tropism in the H1 subtype[21,22]. Nonetheless, neither P227S nor N225G/P227S could compensate D190E in Perth09, Vic11, Mich14, and NDako16 (Supplementary Fig. 3b). Bris07 differed from Perth09 by two substitutions in the RBS region, namely K189N and L194P. Introduction of L194P, but not

K189N, into Perth09 D190E/P227S partially restored virus replication (Supplementary Fig. 3c), suggesting that the partial compensatory effect of P227S is L194P-dependent. Together, these results strongly support that the loss of E190D reversibility is attributable to a complex interconnected series of substitutions.

**Importance of RBS-proximal residues in E190D reversibility.** While G186V reversion was able to restore the reversibility of E190D in Bris07, most H3 strains before early 2000s, including HK68, Shang93, and Mos99, carry a Ser instead of a Val at residue 186. But in contrast to G186V, G186S was not able to exert the same compensatory effect (Fig. 2c), even in the presence of all other substitutions that contributed to reversibility of E190D, namely F193S, P194L, N225G, and P227S. This observation further implies that substitutions at other residues may also be important for the entrenchment of E190D.

To identify additional reversions that can restore the reversibility of E190D, the amino-acid sequence of Bris07 was compared to that of Mos99. In addition to G186S, F193S, P194L, N225G, and P227S, Mos99 differed from Bris07 by nine RBS-proximal reversions (Fig. 3a), which include eight surface

substitutions (T155H, H156Q, F159Y, K160R, N189S, I192T, A196T, R222W) and a buried substitution (I202V). We probed for the fitness effects of all possible combinations of those nine substitutions ($n = 2^9$ variants) in the genetic background of Bris07 G186S/D190E/F193S/P194L/N225G/P227S (rev6). The results from three biological replicates of this deep mutational scanning experiment were strongly correlated (Pearson correlation = 0.89–0.90, Supplementary Fig. 3d). As shown by sequence logo analysis, H156Q, F159Y, and A196T were present in each of the top 10 fittest variants (Fig. 3b), indicating their importance in modulating E190D reversibility. Our virus rescue experiment confirms this notion (Fig. 3c). Addition of H156Q, F159Y, or A196T alone was able to increase the virus titer of Bris07-rev6 by around 2 logs, whereas all three substitutions together increased the virus titer by around 4 logs. A multicycle growth kinetic assay further showed that the growth kinetics of rev6/H156Q/F159Y/A196T, G186V/D190E/F193S/P194L/P227S, and D190E/N225G/P227S mutants are comparable to WT (Fig. 3d). Overall, these results suggest that substitutions in the 150-loop and potentially outside the RBS may also influence the E190D reversibility.

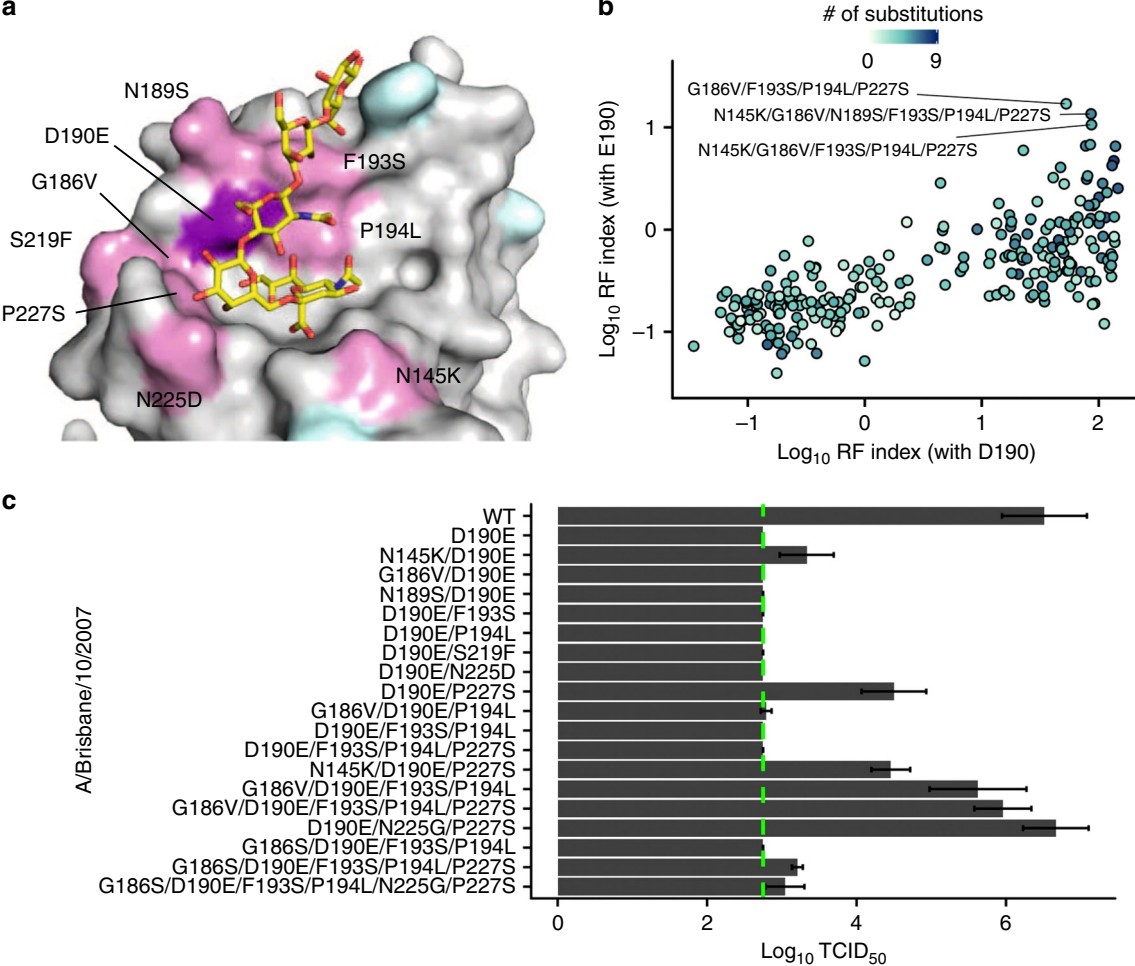

**Fig. 2** Compensatory mutations for the deleterious effect of D190E in Bris07 HA. **a** Substitutions that represent reversions from Bris07 to Wy03 are highlighted in pink or cyan. Pink: residues included in the deep mutational scanning experiment. Cyan: residues not characterized in this study. Purple: residue 190. The receptor analog LS-tetrasaccharide c (LSTc) is colored in yellow. PDB 2YP4 (ref. [9]). **b** Potential compensatory substitutions of Bris07 D190E were probed by deep mutational scanning. Each datapoint represents a certain combination of substitutions. The position along the y-axis represents the $\log_{10}$ RF index for the indicated combination of substitutions plus D190E, whereas position along the x-axis represents the $\log_{10}$ RF index for the indicated combination of substitutions by themselves (without D190E). **c** Effect of different combinations of substitutions on Bris07 replication fitness was examined by virus rescue experiment. Virus titer was measured by TCID$_{50}$. Error bars indicate the standard deviation of three independent experiments. The green dashed line represents the lower detection limit

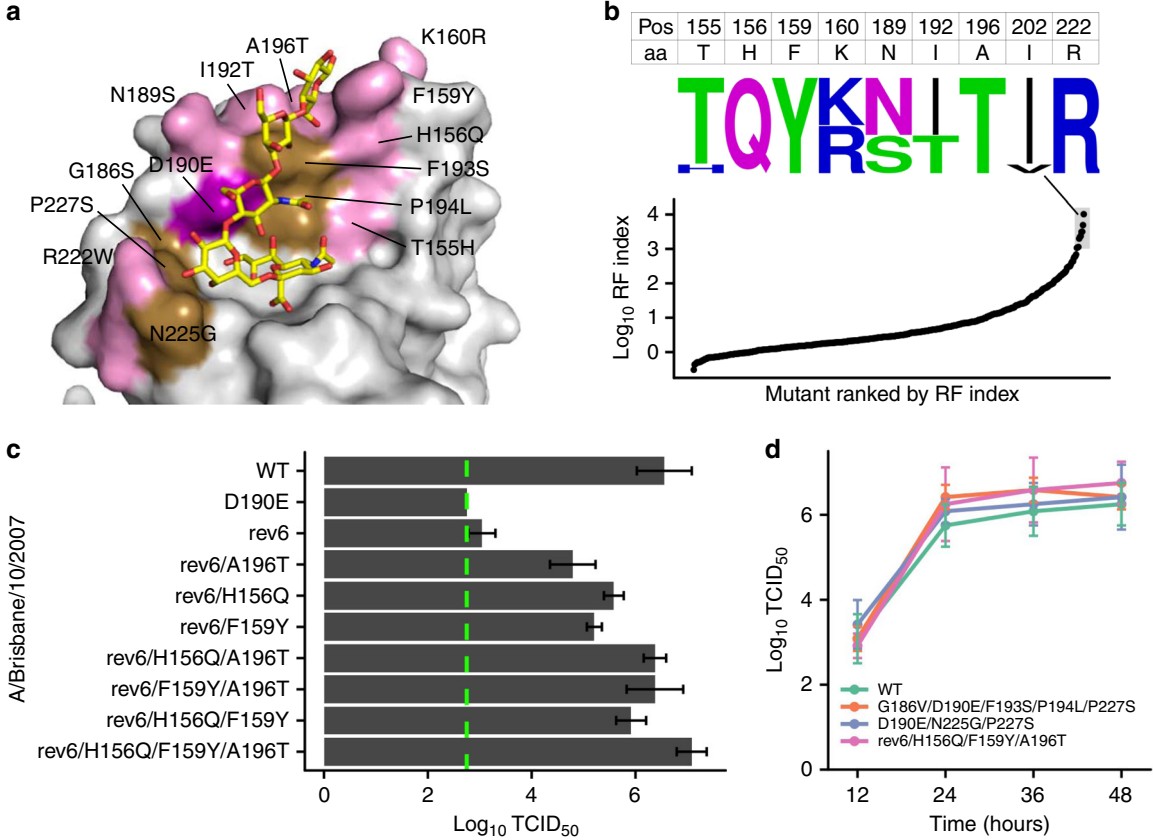

**Fig. 3** Reversibility of E190D is modulated by RBS-proximal residues. **a** Substitutions that represent reversion from Bris07 to Mos99 are highlighted in pink or brown. Pink: residues included in the deep mutational scanning experiment. Of note, a buried residue, namely I202V, is also included in the deep mutational scanning experiment. Brown: residues 186, 193, 194, 225, and 227. Purple: residue 190. The receptor analog LS-tetrasaccharide c (LSTc) is colored in yellow. PDB 2YP4 (ref.[9]). **b** Potential compensatory substitutions of Bris07 G186S/D190E/F193S/P194L/N225G/P227S (rev6) were probed by deep mutational scanning. Each datapoint represents a certain combination of substitutions. The position along the y-axis represents the $\log_{10}$ RF index. All mutants were ranked by the $\log_{10}$ RF index. The amino-acid sequences of the top 10 fittest mutants are represented by the sequence logo. The amino acids (aa) of Bris07 WT are shown above the sequence logo. **c**, **d** Effect of different combinations of substitutions on Bris07 replication fitness was examined by **c** virus rescue experiments and **d** a multicycle growth kinetic assay with a multiplicity of infection (MOI) of 0.01. Virus titer was measured by $TCID_{50}$. Error bars indicate the standard deviation of three independent experiments. The green dashed line represents the lower detection limit. rev6: G186S/D190E/F193S/P194L/N225G/P227S

**Positioning of the 190-helix correlates with E190D reversibility**. We further examined the structural basis for the loss of E190D reversibility in recent H3 strains by determining HA structures of HK68 E190D, Wy03, Wy03 D190E, and Mich14 (Supplementary Table 2). For the purpose of our structural analysis, we also utilized the HA structures from four different human H3 strains, namely HK68, A/Finland/486/2004 (Fin04)[9], A/Hong Kong/4443/2005 (HK05)[9], and Vic11 (ref. [23]). We then compared these structures with HK68 by aligning the receptor-binding subdomain (HA1 residues 117–265)[24]. A slight shift was observed for the 190-helix of the RBS in those H3 strains isolated after year 2003 (Fin04, HK05, Vic11, and Mich14), but not Wy03 (Fig. 4a). This finding coincides with the aforementioned notion that the reversibility of E190D was lost during early to mid-2000s (Fig. 1c). As a result, we considered that the positioning of the 190-helix is also associated with the loss of E190D reversibility.

We therefore analyzed the relative position of the HA 190-helix across different H3 strains by measuring the distance between the Cα of residue 190 ($C\alpha_{190}$) and the phenolic oxygen of residue 98 ($OH_{98}$) (Fig. 4b). The $C\alpha_{190}$-$OH_{98}$ distances in HK68 WT (Glu190), HK68 E190D, Wy03 WT (Asp190), and Wy03 D190E are around 9.0–9.7 Å, whereas this distance in Fin04, HK05, Vic11, and Mich14 is around 8.5–8.8 Å. The $C\alpha_{190}$–$OH_{98}$

distance also correlates well with the $C\alpha_{186}$–$C\alpha_{228}$ distance (Pearson correlation = 0.88; Supplementary Fig. 4a). The $C\alpha_{186}$–$C\alpha_{228}$ distance in HK68 and Wy03 is around 4.5 to 4.9 Å, whereas in Fin04, HK05, Vic11, and Mich14, it is around 4.2 to 4.5 Å (Fig. 4b). The $C\alpha_{186}$–$C\alpha_{228}$ distance can be attributed to the side chain present at residue 186. HK68 has a Ser at residue 186 and Wy03 has a Val, whereas Fin04, HK05, Vic11, and Mich14 have a Gly. The presence of a side chain at residue 186 may increase the $C\alpha_{186}$–$C\alpha_{228}$ distance, as well as the $C\alpha_{190}$–$OH_{98}$ distance, and reposition the 190-helix further from the 220-loop. This notion is consistent with the fact that G186V can restore E190D reversibility (Fig. 2c).

However, G186S did not exert the same D190E compensatory effect as G186V (Fig. 2c), likely due to a slightly smaller side chain. In the presence of G186S, our results from mutagenesis experiments show that H156Q, F159Y, A196T can restore the E190D reversibility (Fig. 3c). These three substitutions reside at the interface of the 150-loop and 190-helix, and are close to each other (Supplementary Fig. 4b). Therefore, it is reasonable to postulate that they also play a role in raising the 190-helix upward to restore the E190D reversibility by modulating the interaction between the 150-loop and 190-helix. Overall, these results show that the structural changes associated with the loss of E190D

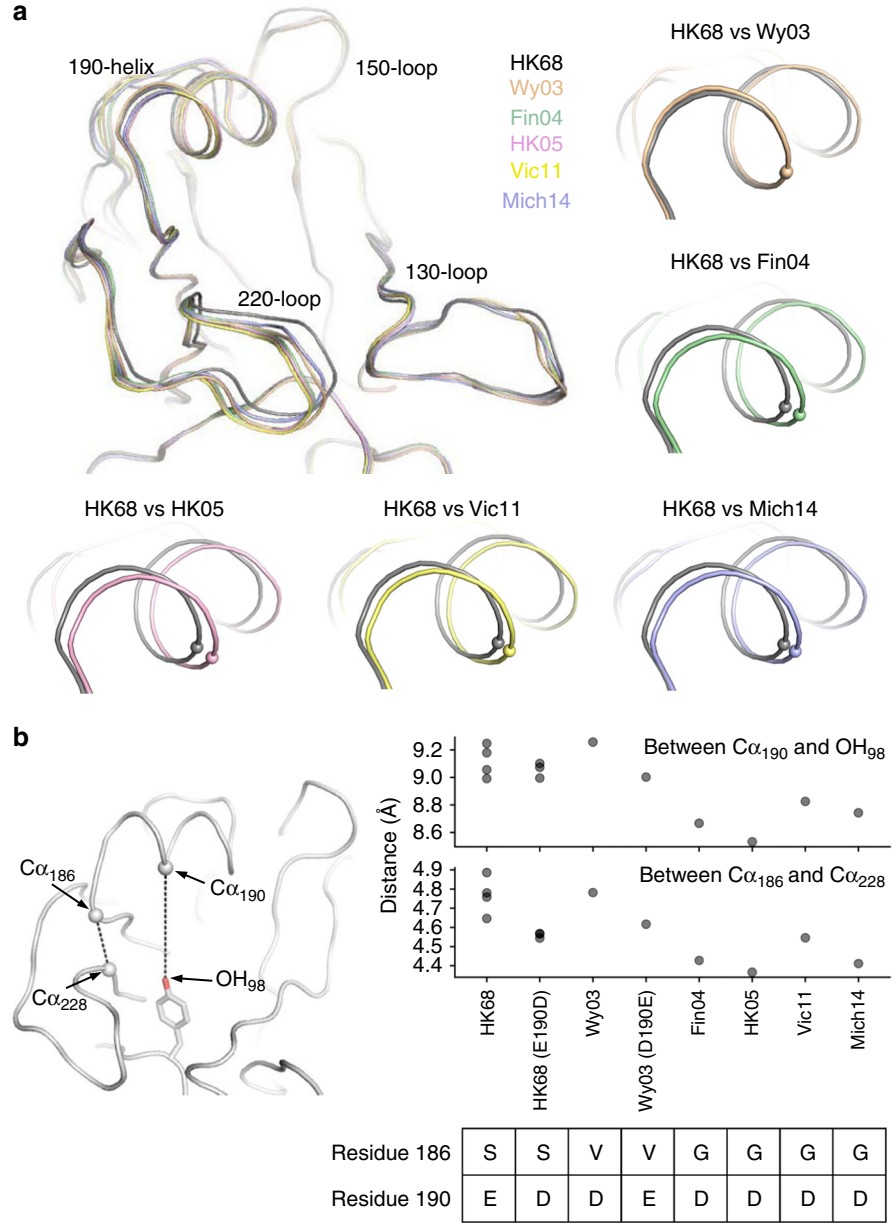

**Fig. 4** Structure comparison of HA RBS from different strains. **a** The HA RBS conformations of HK68 E190D (gray), Wy03 (orange), Fin04 (green), HK05 (pink), Bris07 (blue), and Vic11 (yellow) were compared by aligning their receptor-binding subdomain (HA1 residues 117–265)[24]. Pairwise comparisons of the 190-helix region are also shown. Cα of residue 190 for each HA is shown in a sphere representation. **b** The distances between the Cα of residues 186 (Cα$_{186}$) and the Cα of residues 228 (Cα$_{228}$) and between the Cα of residues 190 (Cα$_{190}$) and the phenolic oxygen of residue 98 (OH$_{98}$) in different strains were measured. HK68: PDB 4FNK (ref. [59]) and PDB 4WE4 (ref. [66]). Fin04: PDB 2YP2 (ref. [9]). HK05: PDB 2YP7 (ref. [9]). Vic11: PDB 4O5N (ref. [23])

reversibility are subtle, but functionally important, and why an epistatic network has evolved.

**Influence of 190-helix positioning on receptor binding.** To understand the mechanistic influence of the 190-helix position on receptor binding, we determined the crystal structure of Wy03 D190E in complex with the human receptor analog 6′-sialyl-*N*-acetyllactosamine (6′-SLN; Supplementary Table 2). The H-bond distance between Glu190 and Sia-1 O9 atom is around 2.5 Å (Fig. 5a), similar to that in HK68, which is around 2.4 Å (Supplementary Fig. 4c)[9]. Both of these values above are at the low end of the H-bond distance range (2.2–3.5 Å). Since any further decrease in this distance may prohibit binding to sialic acid due to steric effects, the lowering of the 190-helix in the HAs of recent H3 viruses could reduce or abolish binding of the D190E

revertant to sialylated glycans. This observation supports that lowering of the 190-helix during the natural evolution of human H3N2 viruses is a major determinant for the E190D reversibility.

**Evolution of receptor binding and E190D reversibility.** Glycan array analysis was performed for WT HK68, Wy03, and Vic11, and the corresponding E190D (HK68) or D190E substitutions. Relative to the WT HAs, the mutants bound more weakly or not at all to the array, but when binding was observed, specificity for human-type (α2–6) receptors was strongly maintained (Fig. 5b). This stronger binding of the WT HAs shows that Glu is preferred at residue 190 of HK68, whereas Asp is preferred in Wy03, even though function is retained. The complete loss of binding to the array observed for Vic11 D190E mutant is consistent with the notion that other substitutions are needed to restore fitness of the

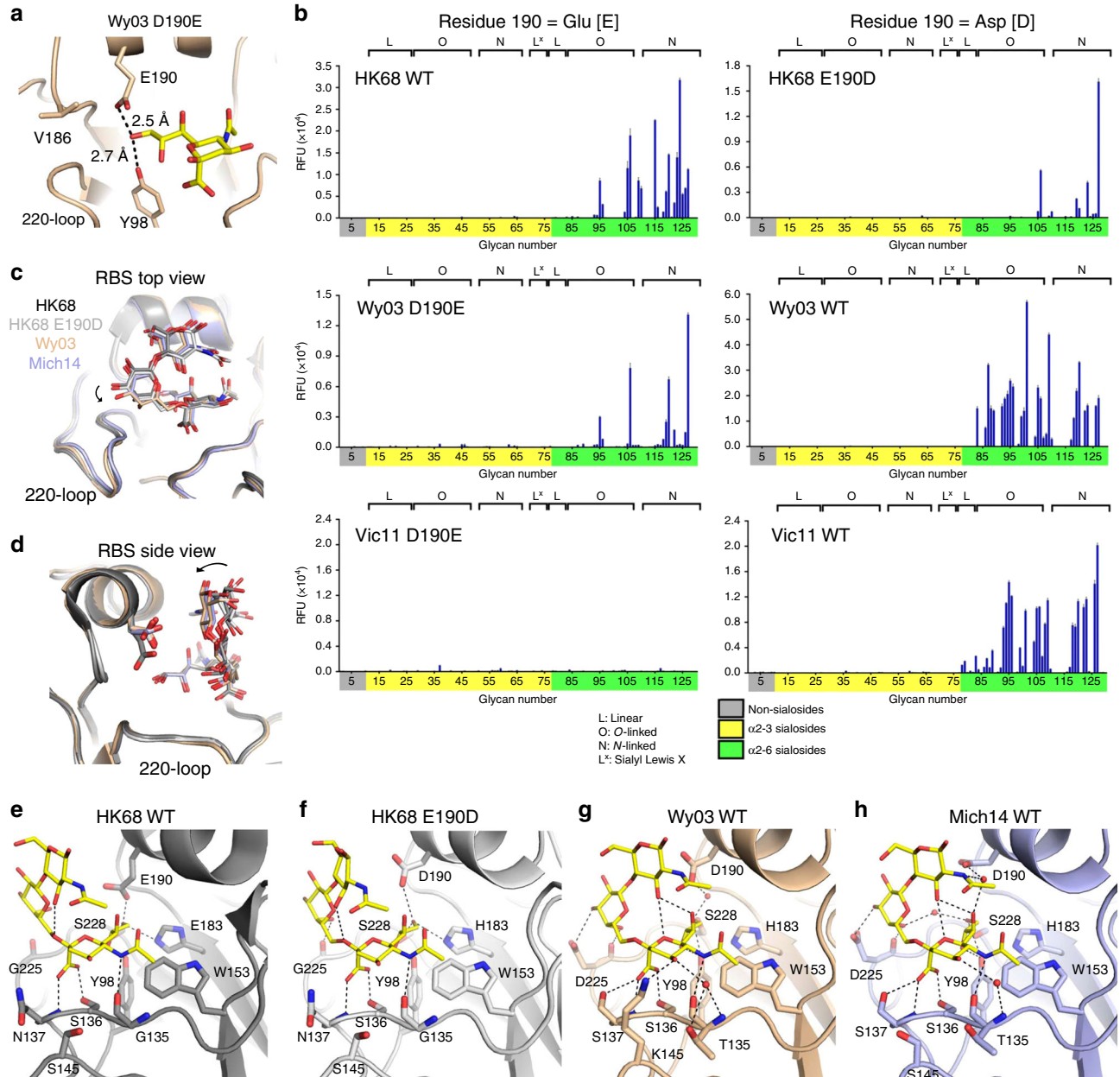

**Fig. 5** Positioning of HA residue 190 in different H3 strains. **a** The interaction between the RBS of Wy03 D190E and sialic acid (yellow sticks) is shown. **b** 293S-expressed recombinant HA was purified and analyzed by sialoside glycan array. HAs in the left column carry a Glu at residue 190. HAs in the right column carry an Asp at residue 190. **c**, **d** The orientations of human receptor analog 6′-SLN in complex with HK68 WT, HK68 E190D, Wy03 WT, and Mich14 WT are compared. The orientation of 6′-SLN in complex with Wy03 WT and Mich14 deviates from that with HK68 WT and HK68 E190D. This deviation is indicated by the curved arrow. **c** Top view of the RBS. **d** Side view of the RBS. **e–h** The interactions of 6′-SLN (yellow sticks) with **e** the RBS of HK68 WT (PDB 2YPG[9]), **f** the RBS of HK68 E190D, **g** the RBS of Wy03 WT, and **h** the RBS of Mich14 WT are shown. Hydrogen bonds are represented by dashed lines

D190E reversion in recent human H3 HAs. Consistently, subsequent analysis of the high-fitness variant Bris07-rev6/H156Q/ F159Y/A196T also showed strong binding to the array (Supplementary Fig. 4d).

To dissect the structural basis of such amino-acid preference at residue 190, we further determined the structures of HK68 E190D, Wy03 WT, and Mich14 WT HAs in complex with human receptor 6′-SLN (Supplementary Table 2). Here, Mich14 is used as a representative of recent human H3 HA. The binding orientations of the human receptor in HK68 WT align well with that in HK68 E190D, but not with those in Wy03 WT and Mich14 WT (Fig. 5c,d). This structural comparison defines two

distinct receptor-binding modes. In fact, the binding orientation of the human receptor analog is very similar among Wy03 WT, Fin04 WT, and Mich14 WT (Supplementary Fig. 5), despite lowering of the 190-helix in Fin04 and Mich14 (Fig. 4b). As compared to HK68 WT and E190D, Gal-2 of the human receptors in Wy03 WT and Mich14 WT moves towards the 220-loop (Fig. 5c) and GlcNAc-3 packs closer to the 190-helix (Fig. 5d). As a result, Asp at residue 190 in Wy03 and Mich14 acquires a van der Waals contact with GlcNAc-3 (distance at around 4 Å). This contact is likely to be important for human receptor binding in Wy03, since D190E weakens the binding to sialylated glycans (Fig. 5b) and the electron density of Gal-2 and

GlcNAc-3 of 6′-SLN in Wy03 D190E is much weaker than that in WT such that they cannot be modeled (Supplementary Figs. 6 and 7). Of note, the E190D substitution in HK68 does not change the binding orientation of 6′-SLN (Fig. 5c,d), suggesting the receptor-binding orientation is not determined by residue 190.

The hydrogen bond (H-bond) network plays a critical role in the evolution of the receptor-binding orientation (Fig. 5e–h). Asp190 in Mich14 makes a water-mediated H-bond with GlcNAc-3 (Fig. 5h). This interaction is absent in HK68 E190D and Wy03 and may also contribute to the preference of Asp over Glu at residue 190 in recent human H3 HAs. The H-bond between Asp225 and Gal-2 in Wy03 (Fig. 5g) and in Mich14 (Fig. 5h) may contribute to the change of receptor-binding orientation from HK68 (Fig. 5c, d), which has Gly at residue 225. This extra H-bond in Wy03 and Mich14 contributes to a twisting of the receptor, where Gal-2 is pulled out from the RBS surface and GlcNAc-3 is in turn rotated towards closer to the 190-helix. This interaction is likely present in other recent H3 HAs that contain an Asn at residue 225 (Fig. 1a and Supplementary Table 1), which is able to form a similar H-bond interaction. This observation provides the structural basis for the D190E compensatory effect of N225G. Nonetheless, we acknowledge that the structural bases for the D190E compensatory effect of other mutations, such as F193S and P227S, are less clear and require further investigation.

In summary, the entrenchment of E190D in recent H3N2 viruses appears to arise in a two-step manner during the natural evolution of human H3N2 viruses. Firstly, structural comparison between HK68 and Wy03 (Fig. 5c–h) suggests that the evolutionary entrenchment initially emerged from a slight change in receptor-binding orientation. Secondly, lowering of 190-helix further entrenches Asp at residue 190 and disfavors Glu. All in all, our mutagenesis and structural analyses strongly suggest that the natural evolution of RBS, which involves the 150-loop, 190-helix, 220-loop and likely the overall shape and charge of the RBS (Supplementary Fig. 8), favors an alternative mode of human receptor binding in recent H3 strains and fixes residue 190 as Asp.

**Residue 190 interacts epistatically with the 220-loop**. Finally, we aimed to explore the potential impact on the evolution of the 220-loop due to E190D substitution. The 220-loop is the least conserved part of the HA RBS and is likely involved in an intragenic epistatic network with residue 190. Epistasis between residues 190 and 225 was involved in the human adaptation of H1 subtype[25], whereas epistasis between residues 226 and 228 is important for the human adaptation of H2 and H3 subtypes[7,26]. In addition, our recent study demonstrated that residues 225, 226, and 228 form a highly epistatic network[15]. These observations indicate that different amino acids at residue 190 might differentially influence the evolution of the 220-loop. Subsequently, a deep mutational scanning experiment was performed on residues 225, 226, and 228 in four different genetic backgrounds of HA, namely Vic11 WT, Vic11 D190E, HK68 WT, and HK68 E190D. Three biological replicates were performed for the deep mutational scanning experiment and their correlations were moderately strong (Pearson correlation = 0.69 to 0.81, Supplementary Fig. 9).

All mutant libraries, including that for Vic11 D190E, were rescued to a virus titer of at least $10^6$ as measured by TCID$_{50}$. Thus, variants that could compensate Vic11 D190E were likely to be present in the mutant library. In fact, Thr at residue 228 was enriched in high-fitness variants in the Vic11 D190E genetic background (Fig. 6a), implying that S228T can compensate for D190E in Vic11. Structural modeling using Rosetta[27,28] further

suggests that S228T substitution shifts the 190-helix upward (Supplementary Fig. 10), reaffirming the positioning of 190-helix as a determinant of E190D reversibility. This compensatory effect of S228T substitution was validated by virus rescue experiments on individually constructed mutants (Fig. 6b). Three different variants that carried a Thr at residue 228, namely S228T, N225G/S228T, and N225G/I226S/S228T, were able to compensate for D190E. But interestingly, these variants were deleterious when introduced into Vic11 WT (with D190), where S228T and N225G/S228T decreased the virus titer by 2 logs and N225G/I226S/S228T by 4 logs. This result shows that the amino-acid preferences in the 220-loop of Vic11 are influenced by the amino-acid identity at residue 190.

Similarly, most variants that were highly fit in the Vic11 D190E genetic background were deleterious in the Vic11 WT genetic background and vice versa, which led to a modest correlation of RF index for each variant between Vic11 WT and D190E genetic backgrounds (Pearson correlation = 0.31, Fig. 6a). However, such a correlation was much stronger in HK68 (Pearson correlation = 0.69, Fig. 6c). Thus, in contrast to Vic11, the amino-acid preferences in the 220-loop of HK68 are largely independent on the amino-acid identity at residue 190. We also observed that the correlation of RF index for each variant between HK68 WT and Vic11 WT genetic backgrounds was relatively modest (Pearson correlation = 0.33, Fig. 6d), implying the amino-acid preferences in the 220-loop differ between HK68 and Vic11. Therefore, the amino-acid preferences in the 220-loop, and how they are modulated by residue 190, have changed over time.

## Discussion

Intragenic epistasis has been empirically shown to constrain the natural evolution of proteins from a variety of viruses and organisms[12,29–36]. Detailed investigation of epistatic interactions often facilitate biochemical and biophysical understanding of the protein of interest, as exemplified by studies of glucocorticoid receptor evolution[12,34], altitude adaptation in hemoglobin[35], drug resistance of β-lactamase[36], and also by our work here. Of note, other qualitative terms, such as suppressor mutations and compensatory mutations[37,38], have also been used to describe the observations in our study and in other similar phenomena.

One major finding in this study is the relationship between the evolution of the receptor binding and the entrenchment of E190D in more recent human H3N2 virus HAs. Our structural and glycan array analyses imply that HAs in early human H3N2 strains favor Glu190 over Asp190 for binding to sialylated glycans, whereas the preference shifted towards Asp190 during natural evolution due to the change in the receptor-binding mode. For HAs in recent human H3 strains, lowering of the 190-helix further entrenched Asp190 such that D190E reversion completely abolishes receptor binding and viral replication. Although our characterization focused on the mechanism and mode of receptor binding, we acknowledge that protein stability could also contribute to the epistasis underlying the entrenchment of E190D (ref. [31]).

The complexity of the intragenic epistatic network identified in this study suggests that the receptor-binding mode of HA is modulated by the coordination of many residues within and adjacent to the RBS. While most RBS substitutions examined in this study naturally emerged after the mid-1990s, many other RBS substitutions arose before that (Fig. 1a). Those uncharacterized substitutions may also have contributed to the change in receptor-binding mode; hence, the shift in amino-acid preference from Glu to Asp at residue 190. It is likely that the amino-acid preference of other residues in the HA RBS has also had to change over time as the receptor-binding mode evolved. In this

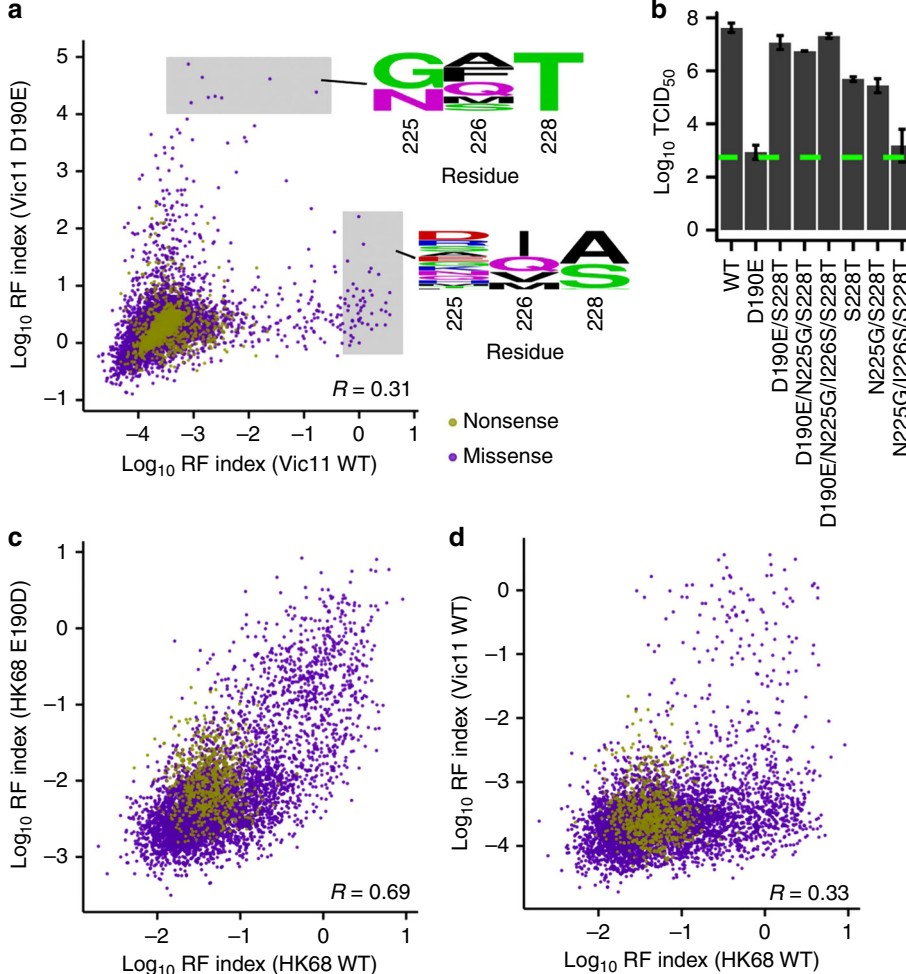

**Fig. 6** Deep mutational scanning of residue 225, 226, and 228. **a** $Log_{10}$ RF index of each variant is compared between Vic11 WT and D190E genetic backgrounds. **b** Effect of different combinations of substitutions on Vic11 replication fitness was examined by virus rescue experiments. Virus titer was measured by $TCID_{50}$. Error bars indicate the standard deviation of three independent experiments. The green dashed line represents the lower detection limit. **c, d** $Log_{10}$ RF index of each variant is compared between **c** HK68 WT and E190D genetic backgrounds and **d** Vic11 WT and HK68 WT genetic backgrounds

work, we used replication in MDCK-SIAT1 cells as a measure of receptor-binding fitness. Nonetheless, evolution in humans is likely to impose other functional constraints on the HA, such as a stronger selection pressure on retaining specificity for human-type receptors. Future studies that explore the constantly evolving sequence-structure relationship of HA-receptor binding are warranted to improve our understanding of the evolutionary constraints on influenza virus.

Interestingly, all substitutions within the epistatic network identified in this study reside in the major antigenic sites[39], including site B (H156Q, F159Y, G186V, D190E, F193S, P194L, A196T), site A (N145K), and site D (N225G and P227S). HA antigenic properties can in fact be altered by substitutions N145K (refs. [40,41]), H156Q (ref. [42]), D190E (refs. [10,43,44]), and P194L (refs. [20,40]). Furthermore, substitutions in residues 159, 193, 196, and 225 have been associated with major antigenic cluster transition[2,45]. Consistently, our results from a hemagglutination inhibition (HAI) assay have demonstrated that the antigenicity of several mutants that promoted the reversibility of E190D, namely rev6/H156Q/F159Y/A196T, G186V/D190E/F193S/P194L/P227S, G186V/D190E/F193S/P194L, and D190E/N225G/P227S, changed compared to WT (Supplementary Fig. 11). Together, these observations strongly suggest that the intragenic epistatic network

identified here is not only relevant to receptor binding, but also a response to immune pressure and subsequent evasion from neutralizing antibodies. It is well known that the HA of human influenza virus evolves in a directional manner[46], where recycling of epitopes is uncommon—antigenic changes rarely revert back to previous sequences. This phenomenon is often attributed to the need for evolving away from the existing immunity. Our study further suggests that entrenchment due to epistasis provides another, non-mutually exclusive, explanation.

It has almost been 50 years since H3N2 influenza viruses began circulating in the human population, and almost 100 years since the 1918 H1N1 pandemic. However, we seem to be far from eradicating or even controlling human influenza viruses due to their rapidly evolving nature. The significant evolutionary capacity of HA can be highlighted by the fact that different influenza A virus subtypes, which shared a common ancestor about 2000 years ago[47], have the same protein fold in their HAs with as low as 31% sequence identity in HA1 and 50% in HA2[48]. The recent discovery of subtypes H17N10 and H18N11 in bats further suggests that HAs from influenza A virus can evolve away from using sialylated glycans as their host receptor[49–51]. Despite the first HA structure being reported more than 35 years ago[52], we still have very limited understanding of its evolvability and the

important consequences on stability, antigenicity and receptor binding. Whether H3N2 viruses will eventually be unable to mutate to escape from the immune system without losing vital functions is still an open question.

## Methods

**Analysis of natural evolution of influenza virus.** Sequences were downloaded from Influenza Research Database (www.fludb.org/)[53]. For the analysis in Fig. 1b, an amino-acid variant that had a change of occurrence frequency from 0 to 95% or above would be classified as a selective sweep. The definition of reversion here is that an amino-acid variant that was present at 95% or above, then dropped to 0%, then increased back again to 95% or above.

**Construction of individual mutants.** Individual mutants for validation experiments were constructed using the QuikChange XL Mutagenesis kit (Stratagene) according to the manufacturer's instructions.

**Virus rescue experiments and measuring virus titer.** Virus rescue experiments were performed based on the A/WSN/33 eight-plasmid reverse genetic system[54]. The pHW2000 eight-plasmid reverse genetic system was a kind gift from Gerd Hobom. In this study, chimeric 6:2 reassortants were employed with the HA and NA ectodomains being replaced by those from H3N2 viruses[15]. The cloning strategy of all HA and NA is the same as described previously[15]. Except for the pilot experiment (Supplementary Fig. 1b) where NA from A/Victoria/361/2011 (Vic11) was used, NA from A/Hong Kong/1/1968 (HK68) was used throughout this study. Transfection was performed in HEK 293T/MDCK-SIAT1 cells (Sigma-Aldrich, catalog number: 05071502-1VL) co-culture (ratio of 6:1) at 60% confluence using lipofactamine 2000 (Life Technologies) according to the manufacturer's instructions. At 24 h post-transfection, cells were washed twice with phosphate-buffered saline (PBS) and cell culture medium was replaced with OPTI-MEM medium supplemented with $0.8\ \mu g\ mL^{-1}$ TPCK-trypsin. Virus was harvested at 72 h post-transfection. For measuring virus titer by $TCID_{50}$ assay, MDCK-SIAT1 cells were washed twice with PBS prior to the addition of virus, and OPTI-MEM medium was supplemented with $0.8\ \mu g\ mL^{-1}$ TPCK-trypsin.

Since HEK 293T cells were able to survive in OPTI-MEM medium supplemented with $0.8\ \mu g\ mL^{-1}$ TPCK-trypsin, virus production, if any, should occur throughout the 72 h post-transfection. In addition, for $TCID_{50}$ assay was performed without changing the cell culture medium after the addition of virus. Therefore, any infectious virus produced in between 24 h post-transfection and 72 h post-transfection should be detected. While we were able to consistently rescue most WT viruses to at least $10^6\ TCID_{50}$, such observations may seem to contradict to Lin et al., which demonstrated that recent human H3N2 strains did not grow well in MDCK-SIAT1 cells[9]. However, these results are not entirely comparable. For instance, the virus strains employed in Lin et al. and this study do not overlap, suggesting that the apparent differences may be due in part to strain-strain variations. In addition, Lin et al. investigated intact H3N2 viral stocks derived from native isolates, whereas we focused on engineered H3 HAs within the constant genetic background of a highly tissue culture-adapted virus (WSN). Therefore, this difference may also contribute to the seemingly discrepant observations between Lin et al. and this study. Notwithstanding, further mechanistic explanations are outside the scope of this study.

**Construction of Bris07 mutant library.** The mutant library was generated by an overlapping PCR strategy. The first fragment of the overlapping PCR was amplified from pHW2000-Bris07-WT (cloned as previously described[15], see above) using primers: 5′-CTG CTT GCA TAA GGA GAT CTA ATA ASA GTT TCT TTA GTA GAT TGA ATT GGT-3′ and 5′-*CRG GRA GAT TTG STC AYT GTC CGT AMC* CGG GTG GTG AAC CCC CCA AAT GT-3′. The second fragment of the overlapping PCR was amplified from pHW2000-Bris07-WT using primers: 5′-*GKT ACG GAC ART GAS CAA ATC TYC CYG* TAT GCT CAA GCA TCA GGA AGA AT-3′ and 5′-CAA TAG ATG CTT ATT CTG CTG GRG ATA TYC CTT ACT CTG GGT CTA RAT CCG ATA TTC GGG-3′. The primers for generating those two fragments described above contained degenerated nucleotides (e.g. S = G or C, R = A or G, Y = C or T, M = A or C, K = G or T). Symbols for degenerate nucleotides followed IUPAC nomenclature. The first fragment randomized amino-acid residues 145, 186, 189, 190, 193, and 194. The second fragment randomized amino-acid residues 186, 189, 190, 193, 194, 219, 225, and 227. The 3′ end of the first fragment overlaps with the 5′ end of the second fragment. Specifically, the italics regions in the above primers were reverse-complemented, allowing subsequent overlapping PCR to assemble the two fragments into a single product. Of note, the overlapping region contained the randomized amino-acid residues 145, 186, 189, 190, 193, and 194. The products of those two PCRs were then mixed at equal molar ratio and used as a template for the overlapping PCR using primers: 5′-CGT ACG TCT CAC TGC TTG CAT AAG GAG ATC TAA-3′ and 5′-CGT ACG TCT CAC AAT AGA TGC TTA TTC TGC-3′. The product of this overlapping PCR was the insert for the mutant library. The vector for the mutant library was generated by a PCR using pHW2000-Bris07-WT as a template and primers: 5′-CGT ACG TCT CAG CAG AGC TTG TTC CGT TTT GAG TGA-3′

and 5′-CGT ACG TCT CAA TTG GAC AAT AGT AAA ACC GGG AGA-3′. Both the vector and insert were digested by *Bsm*BI (New England Biolabs) and ligated using T4 DNA ligase (New England Biolabs). The ligated product was transformed into MegaX DH10B T1R cells (Life Technologies). At least one million colonies were collected. Plasmid mutant libraries were purified from the bacteria colonies using Maxiprep Plasmid Purification (Clontech Laboratories). All PCRs for mutant library constructions were performed using KOD DNA polymerase (EMD Millipore, Billerica, MA) according to the manufacturer's instructions.

**Construction of Bris07-rev6 mutant library.** The mutant library was generated by an overlapping PCR strategy using pHW2000-Bris07-G186S/D190E/F193S/P194L/N225G/P227S (Bris07-rev6) as a template. The forward primer for the first fragment of overlapping PCR was an equal molar mixture of: 5′-TCT TTA GTA GAT TGA ATT GGT TGA CCC AST TAA AAT WCA RAT ACC CAG CAT TGA ACG TG-3′ and 5′-TCT TTA GTA GAT TGA ATT GGT TGC ACC AST TAA AAT WCA RAT ACC CAG CAT TGA ACG TG-3′. The purpose of mixing the two primers was to randomize amino-acid residue 155 to both Thr and His, which could not be achieved using degenerated nucleotide. The reverse primer for the first fragment of overlapping PCR was: 5′-*TCT TCC TGA TGC TTG AGY ATA CAG GGA GRT TTG CTC* AYT GTC CGT ACT CGG GTG GTG-3′. The second fragment of the overlapping PCR was amplified using primers: 5′-*GAG CAA AYC TCC CTG TAT RCT CAA GCA TCA GGA AGA* RTC ACA GTC TCT ACC AAA AGA AG-3′ and 5′-ATT CTG CTG GAG ATA CCC CTT ACC CWG GGT CTA GAT CCG ATA TTC GG-3′. The first fragment randomized residues 155, 156, 159, 160, 189, 192, and 196. The second fragment randomized residues 192, 196, 202, and 222. The 3′ end of the first fragment overlaps with the 5′ end of the second fragment. Specifically, the italics regions in the above primers were reverse-complemented, allowing subsequent overlapping PCR to assemble the two fragments into a single product. Of note, the overlapping region contained the randomized amino-acid residues 192 and 196. The products of the two PCRs were then mixed at equal molar ratio and used as a template for the overlapping PCR using primers: 5′-TCT TTA GTA GAT TGA ATT GGT TG-3′ and 5′-ATT CTG CTG GAG ATA CCC CTT AC-3′. The product of this overlapping PCR was the insert for the mutant library. The vector for the mutant library was generated by a PCR using primers: 5′-TCA ATC TAC TAA GGA AAC TGT TAT TAG ATC TCC TTA TGC-3′ and 5′-TAT CTC CAG CAG AAT AAG CAT CTA TTG GAC AAT AGT AAA-3′. The insert was then cloned to the vector using In-Fusion HD Cloning kit (Clontech Laboratories) according to the manufacturer's instructions. Transformation was performed using MegaX DH10B T1R cells (Life Technologies). At least one million colonies were collected. Plasmid mutant libraries were purified from the bacteria colonies using Maxiprep Plasmid Purification (Clontech Laboratories).

**Construction of HK68 and Vic11 mutant libraries.** The methodology for constructing HK68 E190D, Vic11 WT, and Vic11 D190E mutant libraries was similar to that described previously for our HK68 WT triple mutant library[15]. Vic11 WT and D190E mutant libraries shared the same insert, which was generated by PCR using pHW2000-Vic11-WT (cloned as previously described[15], see above), as a template and primers: 5′-CGT ACG TCT CAG AAT AAG GNN KNN KCC TNN KAG AAT AAG CAT CTA TTG GAC AAT-3′ and 5′-CGT ACG TCT CAT ACA TTT TGG AAT GGT TTG TCA TTG-3′. The insert for HK68 E190D mutant library was generated by PCR using pHW2000-HK68-E190D as a template and primers: 5′-CGT ACG TCT CAG GGT AAG GNN KNN KTC TNN KAG AAT AAG CAT CTA TTG GAC AAT-3′ and 5′-CGT ACG TCT CAT ACG TTT TGA AAG GGC TTG TCA-3′. To generate the vector, pHW2000-HK68-E190D, pHW2000-Vic11-WT, and pHW2000-Vic11-D190E were used as templates for the corresponding mutant libraries. The vector for the HK68 E190D library was generated by PCR using primers: 5′-CGT ACG TCT CAC GTA AAC AAG ATC ACA TAT GGA GCA-3′ and 5′-CGT ACG TCT CAA CCC AGG GTC TGG ACC CGA TAT TCG-3′. Vectors for Vic11 WT and D190E libraries were generated by PCR using primers: 5′-CGT ACG TCT CAT GTA AAC AGG ATC ACA TAC GGG GCC-3′ and 5′-CGT ACG TCT CAA TTC TGG GTC TAG ATC CGA TAT TCG-3′. All vectors and inserts were digested by *Bsm*BI (New England Biolabs) and ligated using T4 DNA ligase (New England Biolabs). The ligated product was transformed into MegaX DH10B T1R cells (Life Technologies, Carlsbad, CA). At least one million colonies were collected. The plasmid mutant libraries were purified from the bacteria colonies using Maxiprep Plasmid Purification (Clontech Laboratories). The HK68 WT mutant library was constructed in our previous study[15].

**Deep mutational scanning experiments.** Virus mutant libraries were rescued in HEK 293T/MDCK-SIAT1 cells co-culture (ratio of 6:1) at 60% confluence in a T225 flask (225 cm²) using lipofactamine 2000 (Life Technologies) according to the manufacturer's instructions. At 24 h post-transfection, cells were washed twice with PBS and cell culture medium was replaced with OPTI-MEM medium supplemented with $0.8\ \mu g\ mL^{-1}$ TPCK-trypsin. Virus was harvested at 72 h post-transfection. Virus was titered by $TCID_{50}$ assay using MDCK-SIAT1 cells and was stored at $-80\ °C$ until used. To passage the virus mutant libraries, MDCK-SIAT1 cells in a T225 flask were washed twice with PBS and infected with a multiplicity of

infection of 0.01 in OPTI-MEM medium supplemented with 0.8 µg mL$^{-1}$ TPCK-trypsin. At 2 h post-infection, infected cells were washed twice with PBS and fresh OPTI-MEM medium supplemented with 0.8 µg mL$^{-1}$ TPCK-trypsin was added to the cells. At 24 h post-infection, supernatant was harvested. Each replicate was transfected and passaged independently. Viral RNA was then extracted from the supernatant using QIAamp Viral RNA Mini Kit (Qiagen Sciences, Germantown, MD). The extracted RNA was then reverse transcribed to cDNA using Superscript III reverse transcriptase (Life Technologies). The plasmid or the cDNA from the post-infection viral mutant libraries was amplified by PCR to add part of the adapter sequence required for Illumina sequencing using the following primers:

For Bris07 and Bris07-rev6 mutant libraries: 5′-CAC TCT TTC CCT ACA CGA CGC TCT TCC GAT CTA GCT CTG CTT GCA TAA GGA GAT-3′ and 5′-GAC TGG AGT TCA GAC GTG TGC TCT TCC GAT CTT TGT CCA ATA GAT GCT TAT TCT-3′.

For HK68 WT and E190D mutant libraries:

5′-CAC TCT TTC CCT ACA CGA CGC TCT TCC GAT CTG GGT TCA CCA CCC GAG CAC GAA-3′ and 5′-GAC TGG AGT TCA GAC GTG TGC TCT TCC GAT CTA ACT ATT GTC CAA TAG ATG CTT-3′.

For Vic11 WT and D190E mutant libraries:

5′-CAC TCT TTC CCT ACA CGA CGC TCT TCC GAT CTG GGT TCA CCA CCC GGG TAC GGA-3′ and 5′-GAC TGG AGT TCA GAC GTG TGC TCT TCC GAT CTT ACT ATT GTC CAA TAG ATG CTT-3′.

A second PCR was performed to add the rest of the adapter sequence and index to the amplicon using primers: 5′-AAT GAT ACG GCG ACC ACC GAG ATC TAC ACT CTT TCC CTA CAC GAC GCT-3′ and 5′-CAA GCA GAA GAC GGC ATA CGA GAT XXX XXX GTG ACT GGA GTT CAG ACG TGT GCT-3′. Positions annotated by an "X" represented the nucleotides for the index sequence. The index sequence for each sample is shown in Supplementary Table 3. The final PCR products were submitted for next-generation sequencing. One lane of Illumina MiSeq PE300 was used for both Vic11 and HK68 libraries, 6% of a lane of Illumina MiSeq PE300 was used for Bris07 libraries, and 10% of a lane of Illumina MiSeq PE300 was used for Bris07-rev6 libraries.

**Sequencing data analysis**. Sequencing data were obtained in FASTQ format and were parsed using SeqIO module in BioPython[55]. A paired-end read was filtered and removed if the corresponding forward and reverse reads were not reverse-complemented at each randomized codon. Each paired-end read was then translated. Each variant (at the amino-acid level) was called by comparing individual translated paired-end reads to the WT reference sequence. The number of each variant in each sample then was counted. Subsequently, a relative fitness index (RF index) was computed for each variant:

$$\text{RF index}_{\text{mutant}} = \frac{(\text{Count}_{\text{mutant,post-selection}} + 1)/(\text{Count}_{\text{mutant,input}} + 1)}{(\text{Count}_{\text{WT,post-selection}} + 1)/(\text{Count}_{\text{WT,input}} + 1)}$$

For a given mutant, Count$_{\text{mutant,post-selection}}$ represents the number of paired-end reads in post-selection mutant library, whereas Count$_{\text{mutant,input}}$ represents the number of paired-end reads in the input plasmid mutant library. Thus, the RF index$_{\text{mutant}}$ represents the occurrence frequency change of a given mutant during the selection process relative to that of the WT. A pseudocount was added to all count items to prevent the division by zero. For the analysis of HK68 WT, HK68 E190D, Vic11 WT, and Vic11 D190E libraries, variants with a read count of less than 20 in the input plasmid mutant library were discarded. The RF index for individual variants is listed in Supplementary Data 1.

**Crystallization and structural determination**. HAs were expressed and purified as described for Vic11 HA[56] and were concentrated to 10 mg mL$^{-1}$. Initial crystal screening was carried out using our high-throughput, robotic CrystalMation system (Rigaku, Carlsbad, CA) at TSRI. The initial crystal screening was based on the sitting drop vapor diffusion method at 4 °C and 20 °C with 35 µL reservoir solution and each drop consisting 0.1 µL protein +0.1 µL precipitant. Further refinement of the conditions obtained from the initial hits was performed manually by the sitting drop vapor diffusion method with 500 µL reservoir solution and each drop consisting 0.8 µL protein + 0.8 µL precipitant. To generate HA-receptor complexes, crystals were soaked in reservoir solution supplemented with 20 mM of 6′-SLN for 2 h. The resulting crystals were flash cooled and stored in liquid nitrogen until data collection. The crystallization conditions were as follows:

A/Hong Kong/1/1968 (HK68) E190D: 0.1 M sodium cacodylate pH 6.5, 6% PEG 8000, and 39% 2-methyl-2,4-pentanediol at 20 °C.

A/Wyoming/3/2003 (Wy03) WT and D190E: 0.1 M bicine pH 9 and 42% 2-methyl-2,4-pentanediol at 20 °C.

A/Michigan/15/2014 (Mich14) WT: 0.1 M Tris pH 8.5, 0.2 M lithium sulfate, and 40% PEG 300 at 4 °C.

Diffraction data were collected at the APS GM/CA-CAT 23ID-B, 23ID-D, and at the Stanford Synchrotron Radiation Lightsource beamline 12-2. The data were indexed, and integrated and scaled using HKL2000 (HKL Research)[57]. The structure was solved by molecular replacement using Phaser[58] with PDB: 4FNK[59] as the molecular replacement model, the structure was rebuilt using Coot[60], and refined using Refmac5 (ref. [61]). Ramachandran statistics were calculated using MolProbity[62]. Electrostatics calculation was performed using PDB2PQR Server[63] using Amber forcefield[64].

**Glycan array analysis**. Glycan array analysis was performed using an NHS ester-coated glass microarray slide containing six replicates of 130 diverse sialic acid-containing glycans, including terminal sequences and intact N-linked and O-linked glycans found on mammalian and avian glycoproteins and glycolipids[65]. Recombinant HAs (50 µg mL$^{-1}$ final) were expressed and purified as described previously[65], before precomplexing with monoclonal mouse anti-His-Tag IgG (MA1-21315; Thermo Fisher Scientific) and goat anti-mouse IgG-Alex Fluor 647 (A-21235; Thermo Fisher Scientific) antibodies for 30 min at RT in 1× PBS-T. HA-antibody complexes were incubated on the array surface for 1 h at room temperature. Slide scanning to detect bound HA was conducted using an Innoscan-1100AL (Innopsys) fluorescent microarray scanner. Fluorescent signal intensity was measured using Mapix (Innopsys) and mean intensity minus mean background of 4 replicate spots was calculated. A complete list of the glycans on the array is presented in Supplementary Fig. 12.

**HAI assay**. Normal ferret sera (FR282), ferret polyclonal sera raised against both native Bris07 H3N2 virus (FR389) and recombinant Bris07 HA (FR289), plus four monoclonal mouse antibodies to recombinant Bris07 HA (FR509-512) were obtained from the International Reagent Resource (IRR). All sera stocks were thawed and pre-treated with 500 units of a mixed neuraminidase cocktail (New England Biolabs, P0720 and P0722) at 37 °C for 3 h with constant mixing, followed by heat inactivation at 56 °C for 30 min. Inactivated sera stocks were diluted 1:5 and antibodies to 20 µg mL$^{-1}$, before making two-fold serial dilution across 96-well U-bottom plates (final serum titer range = 10–20,480, final antibody concentration = 10–0.005 µg mL$^{-1}$). Pre-titered Bris07 WT and mutant stocks were applied to each well at 4 HAU (final) and incubated for 30 min at room temperature. Hemagglutination inhibition was analyzed through addition of 0.5% (final) Turkey red blood cells (Lampire Biological Laboratories), and titers read after a 45-min incubation at room temperature. All titers were calculated in duplicate (due to limited sera availability) and values averaged.

**Code availability**. Custom python scripts for analyzing the deep mutational scanning data have been deposited to https://github.com/wchnicholas/H3Entrench.

**Data availability**. Raw sequencing data have been submitted to the NIH Short Read Archive under accession number: BioProject PRJNA377321. The X-ray coordinates and structure factors have been deposited to the RCSB Protein Data Bank under accession codes: 6BKM, 6BKN, 6BKO, 6BKP, 6BKQ, 6BKR, 6BKS, and 6BKT. All of the other data that support the conclusions of the study are available from the corresponding author upon request.

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

## Acknowledgements
We thank Gerd Hobom for the influenza A/WSN/33 eight-plasmid reverse genetic system, Steven Head, Jessica Ledesma, and Lana Schaffer at TSRI Next Generation Sequencing Core, and Steffen Bernard for assistance with X-ray crystallography data processing. We acknowledge NIH R56 AI117675, R56 AI127371, and R01 AI114730 for support. N.C.W. was supported by a Croucher Foundation Fellowship. A.J.T. is an EMBO Long-Term fellow (ALTF 963-2014).

## Author contributions
N.C.W., A.J.T., J.X., J.C.P., R.A.L., and I.A.W. conceived and designed the study; N.C.W. and J.X. performed the mutagenesis experiments; N.C.W. wrote the computational scripts for data analysis; C.W.L. assisted with the pilot experiment; C.M.N. synthesized

the 6′-SLN; N.C.W. and X.Z. collected the X-ray data; and N.C.W. determined and refined the X-ray structures. A.J.T. performed the sialoside glycan array experiment and hemagglutination inhibition assay. N.C.W. and I.A.W. wrote the paper and all authors reviewed and edited the paper.

## Additional information

**Competing interests:** The authors declare no competing interests.

