## [Peer Review File(PDF 125 kb) · Nature Communications]

Editorial Note: this manuscript has been previously reviewed at another journal that is not operating a transparent peer review scheme. This document only contains reviewer comments and rebuttal letters for versions considered at *Nature Communications*.

Reviewers' comments:

Reviewer #1 (Remarks to the Author):

The authors responded appropriately to most of this reviewer's comments. However, the authors refused to rewrite the manuscript without the evolutionary context. The authors justify this by stating "Entrenchment or irreversibility, which is the central focus of our study, is an important concept in evolutionary biology". This is fine. However, what the authors studied is the incompatibility of the HAD190E mutation in the HA of post-2007 viruses and the identification of the compensatory mutations. This approach does not help our understanding of the evolution of H3 HAs. As I stated in my previous review, the data presented in the manuscript still advance our understanding of the HA structure; however, the context in which the data are presented is not appropriate and, therefore, not suitable for Nature Communications.

Reviewer #3 (Remarks to the Author):

The authors addressed all of my concerns from my earlier review of the manuscript. The manuscript nicely use a combination of deep mutational scanning, site directed mutagenesis, and crystal structures to identify epistatic networks that affect the HA receptor binding site of influenza viruses.

Reviewers' comments:

Reviewer #1 (Remarks to the Author):

The authors responded appropriately to most of this reviewer's comments. However, the authors refused to rewrite the manuscript without the evolutionary context. The authors justify this by stating "Entrenchment or irreversibility, which is the central focus of our study, is an important concept in evolutionary biology". This is fine. However, what the authors studied is the incompatibility of the HAD190E mutation in the HA of post-2007 viruses and the identification of the compensatory mutations. This approach does not help our understanding of the evolution of H3 HAs. As I stated in my previous review, the data presented in the manuscript still advance our understanding of the HA structure; however, the context in which the data are presented is not appropriate and, therefore, not suitable for Nature Communications.

Response: We are encouraged to see that our previous responses addressed most of comments. While reviewer #1 agrees that entrenchment or irreversibility is an important concept in evolutionary biology, reviewer #1 also states that our study does not help our understanding of the evolution of H3 HAs because we are focusing on the incompatibility of the HA D190E mutation in the HA of post-2007 viruses and the identification of the compensatory mutations. However, we feel that the incompatibility of HA D190E exactly demonstrates the irreversibility of the E190D mutation that first arose in circulating H3N2 viruses during 1993. In fact, the meaning of "D190E incompatibility" in our manuscript is equivalent to "irreversibility of E190D". We think using the word "incompatibility" instead of "irreversibility" possibly caused some unnecessary confusion and we have revised the manuscript by replacing the word "incompatibility" with "irreversibility".

E190D mutation is a natural mutation that has important consequences for the virus as it is a key residue in the receptor-binding site for receptor specificity as well as for immune escape. We explore that the consequences of that natural mutation by assessing whether it can be reverted. As shown in Fig. 1c (and also in Supplementary Fig. 1b) of our manuscript, HA E190D first arose in 1993 but was still reversible. However, we found that this reversibility of HA E190D was lost after 2007 and our data clearly show that HA D190 in post-2007 viruses cannot be reverted back to HA E190, which is universal in pre-1993 ancestral strains. This observation parallels a classic paper studying why there can be irreversibility of mutations (Bridgham et al., Nature 2009, PMID: 19779450).

Bridgham et al. found that reverting the glucocorticoid receptor to an ancestral state at some residues (the ancestral sequence was 37 amino acid mutations distant) totally abolishes binding to its ligand. That finding is parallel to our observation of incompatibility of HA D190E in post-2007 viruses. As in our study, Bridgham *et al.* identified compensatory mutations by examining other reversions. The philosophy of their approach is identical to ours. In Bridgham *et al.*, the entire paper was presented in the context of evolution. Bridgham et al. was actually cited as an example of "Dollo's law of irreversibility" in evolutionary biology on Wikipedia (https://en.wikipedia.org/wiki/Dollo%27s_law_of_irreversibility):

"A 2009 study on the evolution of protein structure proposed a new mechanism for Dollo's law. It examined a hormone receptor that had evolved from an ancestral protein that was able to bind two hormones to a new protein that was

specific for a single hormone. This change was produced by two amino acid substitutions, which prevent binding of the second hormone. However, several other changes subsequently occurred, which were selectively neutral as they did not affect hormone binding. When the authors tried to revert the protein back to its ancestral state by mutating the two "binding residues", they found the other changes had destabilised the ancestral state of the protein. They concluded that in order for this protein to evolve in reverse and regain its ability to bind two hormones, several independent neutral mutations would have to occur purely by chance with no selection pressure. As this is extremely unlikely, it may explain why evolution tends to run in one direction."

The case cited above corresponds almost directly to what we have outlined for evolutionary changes in the receptor binding site in influenza hemagglutinin. That being said, we understand the reviewer's statement that this D190E reversion has never occurred in nature. **Therefore, we have revised most subsections of the results section to strictly describe the irreversibility of E190D without the evolutionary implications.**

However, we do believe that our crystal structures directly demonstrate a change in HA receptor-binding mode during the course of human H3N2 evolution. Consequently, we think it is appropriate to retain the evolutionary context in the subsections that present and compare the different HA structures.

Reviewer #3 (Remarks to the Author):

The authors addressed all of my concerns from my earlier review of the manuscript. The manuscript nicely use a combination of deep mutational scanning, site directed mutagenesis, and crystal structures to identify epistatic networks that affect the HA receptor binding site of influenza viruses.

Response: We are pleased to learn that our previous revision addressed all of the concerns from reviewer #3.